

# Risk factors associated with prolonged viral clearance in patients with a refractory course of COVID-19: a retrospective study

Weitao Zhuang[1,2,*], Shujie Huang[1,2,*], Dongya Wang[3,4], Lulu Zha[4,5], Wei Xu[6] and Guibin Qiao[1]

[1] Department of Thoracic Surgery, Guangdong Provincial People's Hospital, Guangzhou, Guangdong, China
[2] Shantou University Medical College, Shantou, Guangdong, China
[3] Department of Hyperbaric Oxygen and Rehabilitation, General Hospital of Southern Theater Command, Chinese People's Liberation Army, Guangzhou, Guangdong, China
[4] The Fifth Department of Infectious Diseases, Huoshenshan Hospital, Wuhan, China
[5] Department of Nursing, General Hospital of Southern Theater Command, Chinese People's Liberation Army, Guangzhou, China
[6] School of Public Health and Management, Chongqing Medical University, Chongqing, China
[*] These authors contributed equally to this work.

Corresponding author
Guibin Qiao, qiaoguibin@gdph.org.cn, guibinqiao@126.com

## ABSTRACT

**Background**. This study aimed to characterize a cohort of patients with a refractory course of COVID-19, and to investigate factors associated with the duration of viral clearance (DoVC).

**Materials & Methods**. A total of 65 patients with refractory COVID-19 were retrospectively enrolled from Huoshenshan Hospital. Univariate analysis and multivariate analysis were performed to examine the potential association between clinicopathologic characteristics and the DoVC.

**Results**. The median DoVC in the overall study cohort was 48 days (ranged from 21 to 104 days). Multivariate analysis indicated that fever at illness onset (Hazard ratio (HR) = 4.897, 95% CI [1.809–13.260], $p = 0.002$), serum level of aspartate aminotransferase (AST) > 21.8 IU/L (HR = 3.010, 95% CI [1.195–7.578], p = 0.019), and titer of SARS-CoV-2 IgG > 142.09 AU/ml (HR = 3.061, 95% CI [1.263–7.415], $p = 0.013$) were the three independent risk factors associated with delayed viral clearance.

**Conclusion**. The current study suggested that clinical characteristics such as fever at illness onset, a high serum level of AST or SARS-CoV-2 IgG were associated with delayed viral clearance. Patients with these characteristics might need a more individualized treatment strategy to accelerate their recovery from the refractory COVID-19.

## INTRODUCTION

As the severe acute respiratory syndrome coronavirus 2 (SARS-CoV-2) sweeps across the globe, there is rising concern that the patients who recovered from COVID-19 may be at risk of reinfection. It was reported that a small percentage of patients were tested positive

again after recovery from a previous episode of SARS-CoV-2 infection (*Mei et al., 2020*; *Smith, 2020*). However, the mechanism of reinfection/reactivation has not been clarified, which created an uncertainty of the criteria to define clinical and virological recovery (*Alvarez-Moreno & Rodriguez-Morales, 2020*). Miscellaneous hypotheses are coming out to address this scientific question, such as false positive results from the leftover genetic materials (*Chaturvedi et al., 2020*; *Batisse et al., 2020*), persistence of virus within body, cross-contamination from another virus, incorrect sampling methods (*Law, Leung & Xu, 2020*), thermal inactivation of samples (*Pan et al., 2020*), impaired immune response (*Batisse et al., 2020*), even storage of viruses in exosomes and extracellular vesicles (*Elrashdy et al., 2020*). Given the false negative rates were as high as 30%–60% for SARS-CoV-2 PCR testings in upper respiratory tract or sputum specimens (*Wang et al., 2020*; *Li et al., 2020*), it is not easy to distinguish between reinfection and prolonged viral clearance (*Man et al., 2020*; *Xiao, Tong & Zhang, 2020*). Underlying mechanisms and corresponding patient characteristics of reinfection or prolonged viral clearance may be quite different, which requires personalized treatment strategy. As the medical institutions are overwhelmed by the cumulating patients, what kind of care should be provided to the reinfected patients, and how to deal with the patients with a refractory course of disease, remain to be answered by both physicians and researchers.

The median duration of viral shedding was reported to be about 20 days in many studies (*Zhou et al., 2020*; *Yan et al., 2020*; *Qi et al., 2020*; *Fu et al., 2020*; *Xu et al., 2020*). However, prolonged viral clearance is not a rare phenomenon, with the longest duration being 83 days (*Li, Wang & Lv, 2020*; *Yang et al., 2020*; *Liu et al., 2020a*). In this regard, a longer period of post-discharged quarantine and medical observation has been suggested (*Mei et al., 2020*; *Li, Wang & Lv, 2020*; *Liu et al., 2020a*). Identifying the patients who are at risk of reinfection or delayed viral clearance might be the first important step to tackle the abovementioned challenge. This may help with optimizing the treatment and preventing the transmission of SARS-CoV-2. Recently, studies are emerging to identify the risk factors associated with longer viral clearance (*Yan et al., 2020*; *Qi et al., 2020*; *Xu et al., 2020*; *Hu et al., 2020*). However, risk factors of prolonged duration of viral clearance (DoVC) in patients with a refractory course of COVID-19 are less clear and deserve further investigation. Mo, et al. defined a hospital stay of >10 days without alleviation of symptoms or radiological improvement to be the refractory COVID-19, which was the first study to investigate the clinical characteristics associated with refractory COVID-19 pneumonia (*Mo et al., 2020*). By their definition, nearly a half of the patients in this study had a refractory course of COVID-19, which suggested a necessity of further stratification (*Mo et al., 2020*).

In our study, we characterized a cohort of patients who were deemed having a refractory course of COVID-19, with the purpose to identify the clinicopathologic risk factors associated with the delayed viral clearance.

## PATIENTS AND METHODS

### Participants

A total of 65 inpatients with refractory COVID-19 from Huoshenshan Hospital between February 8, 2020 to March 31, 2020 were enrolled in this retrospective study. All patients

had a calculated DoVC longer than 20 days and had a previous history of COVID-19-related hospitalization before admitted to Huoshenshan Hospital. Patients with a calculated DoVC ≤ 20 days were excluded from this study. The severity of disease was evaluated based on the sixth version of Chinese clinical guideline for COVID-19 (*National Health Commission of China, 2020*). Mild cases referred to mild clinical symptoms with no sign of pneumonia on imaging. General cases were defined as positive clinical symptoms along with pneumonia on thoracic imaging. Patients who met any of the following criteria were classified as severe cases: respiratory rate ≥ 30 per minute; pulse oxygen saturation ≤ 93% at resting status; arterial partial pressure of oxygen (PaO2)/oxygen concentration (FiO2) ≤ 300 mmHg; patients with >50% progression of lesion volume within 24 to 48 h on chest imaging. Patients who presented with any of the following conditions were defined as critical cases: respiratory failure that requires mechanical ventilation; shock; organ failure that requires ICU care. Patients could be discharged after they met the following criteria (*Zhou et al., 2020*): absence of fever for at least 3 days, remission of clinical symptoms or chest imaging, and two consecutive negative results for SARS-CoV-2 RNA in respiratory tract samples obtained at least 24 h apart. This study was approved by the institutional review boards of Huoshenshan Hospital (No. IEC-AF-SL-02), Guangdong Provincial People's Hospital (No. GDRE20200141H) and General Hospital of Southern Theater Command (No. 202039), and was carried out in accordance with the Declaration of Helsinki. Electronic forms of informed consents to use their demographic and clinical information were collected from all participants in advance.

## Data collection

A standardized sheet for data collection was used to retrieve the demographic, clinical, laboratory and treatment information from electronic medical records. Given the relatively high rate of false negative nucleic acid testing by nasal/pharyngeal swabs in the previous studies (*Wang et al., 2020*; *Li et al., 2020*), we raised the standard to define viral clearance, which was in accordance with the abovementioned standard criteria of discharge from hospital (*Zhou et al., 2020*). Therefore, the DoVC was calculated from the date of illness onset (as stated in the history of present illness) to the date of discharge from Huoshenshan hospital. For those who received their negative results of nucleic acid testing but were delayed for discharge due to complications of other organ systems, the DoVC was calculated from the date of illness onset to the date of the second negative nucleic acid testing. Patients with a DoVC >20 days were defined to have a refractory course of COVID-19 and satisfied the inclusion criteria of this study. All data were double-checked by two researchers (W. Z. and S. H.) independently for accuracy.

## Laboratory tests

Respiratory samples were tested for SARS-CoV-2 RNA by the real-time reverse transcriptase polymerase chain-reaction (RT-PCR) test (BioGerm, Shanghai, China), following WHO guideline for qRT-PCR (*Corman et al., 2020*; *World Health Organization, 2020*). SARS-CoV-2 IgM and IgG antibodies in blood samples were measured using chemiluminescent immunoassay (Shenzhen Yahuilong Biotechnology Co., Ltd) as per manufacturer's

protocol. The titers higher than 10 AU/ml of either IgM or IgG were considered positive infection for SARS-CoV-2.

## Treatment

All patients were provided with necessary supportive care during the first and second hospitalizations, including but not limited to oxygen, antipyretic, antitussive, expectorant, antidiabetic, and antihypertensive agents. For etiological treatment, antivirals, antibiotics, traditional Chinese herbal tea or Chinese patent medicine were used either separately or in combination.

## Statistical analysis

Data of baseline characteristics such as demographic information, reasons of admission, symptoms and signs, comorbidities and treatment details were presented as frequency (percentage), while data of laboratory results were presented as median (interquartile range, IQR). The medians of continuous variables were used as cut-off values to separate the patients into two groups for the sake of comparison, except for C-reactive protein and procalcitonin, where the upper limits of normal range were used. Univariate analysis was performed by using log-rank test to evaluate the association of patient characteristics with DoVC. All variables with $p$ values $\leq 0.1$ were included in multivariate analysis. Subsequently, log minus log function was used to identify the covariates which did not satisfy the proportional hazard assumption, and these covariates were excluded for the subsequent multivariate analysis. Multivariate analysis was conducted using Cox regression model to identify independent risk factors of prolonged viral clearance. Cases with missing data were excluded from the Cox regression model. A sensitivity analysis was performed to assess the robustness of the results (see Tables S1, S2). Finally, Kaplan–Meier method was used to visualize the impact of these risk factors on the DoVC. All data were performed using IBM SPSS statistics for Windows, Version 25.0 (Armonk, NY: IBM Corp., USA). Figures were generated using the R software version 3.6.3 (R Foundation for Statistical Computing, Vienna, Austria).

# RESULTS

## Baseline characteristics

The mean age of patients with a refractory course of COVID-19 was 52.5 years old, and 40 out of 65 patients were male. According to the sixth version of Chinese clinical guideline for COVID-19 (National Health Commission of China, 2020), 51 out of 68 (78.5%) patients were classified as general cases, while the others were severe cases. The number of patients who were admitted because of recurrence of symptoms, possible reinfection with positive nucleic acid testing again or referral with persistent symptoms was 3 (4.6%), 11 (16.9%) and 51 (78.5%), respectively. Fever (78.5%), cough (70.8%), shortness of breath (43.1%) and fatigue (33.8%) were the four most prevalent sign and symptoms at illness onset among these patients. Hypertension (26.2%) was the most common comorbidity, followed by type II diabetes mellitus (18.5%) (Table 1).
**Table 1** Baseline characteristics of patients with a refractory clinical course of COVID-19.

| Baseline Characteristics | Study cohort = 65 frequency (%)/Mean ± SD | p value[*] |
|---|---|---|
| **Age (years)** | 52.5 ± 14.1 | 0.85 |
| ≤60 | 43 (66.2) | |
| >60 | 22 (33.8) | |
| **Male** | 40 (61.5) | 0.76 |
| **Reasons of admission** | | 0.89 |
| Recurrence of symptoms | 3 (4.6) | |
| Possible reinfection with positive nucleic acid testing again | 11 (16.9) | |
| Referral with persistent symptoms | 51 (78.5) | |
| **Symptoms and signs at illness onset** | | |
| Fever | 51 (78.5) | **0.03** |
| Shortness of breath | 28 (43.1) | **0.03** |
| Dyspnea | 5 (7.7) | 0.17 |
| Chest tightness | 11 (16.9) | 0.77 |
| Myalgia | 13 (20.0) | 0.97 |
| Dry cough | 16 (24.6) | 0.70 |
| Productive cough | 30 (46.2) | 0.82 |
| Fatigue | 22 (33.8) | **0.03** |
| Diarrhea | 4 (6.2) | 0.99 |
| Headache | 3 (4.6) | 0.99 |
| **Comorbidities** | | |
| Hypertension | 17 (26.2) | 0.35 |
| Type II diabetes mellitus | 12 (18.5) | 0.91 |
| Coronary artery disease | 4 (6.2) | 0.24 |
| COPD | 3 (4.6) | 0.94 |
| Hepatitis B | 3 (4.6) | 0.50 |
| Malignancy | 2 (3.1) | 0.67 |
| **Smoker** | 5 (7.7) | 0.80 |
| **Alcohol user** | 2 (3.1) | 0.99 |
| **Severity of COVID-19 pneumonia** | | 0.40 |
| Mild | 0 | |
| General | 51 (78.5) | |
| Severe | 14 (21.5) | |
| Critical | 0 | |

**Notes.**
*Univariate analysis to determine the association of baseline characteristics with duration of viral clearance by using log-rank test. COPD, chronic obstructive pulmonary diseases. Variables with p value in bold were included in multivariable analysis.

## Clinicopathologic information

Complete blood count and blood biochemistries such as inflammatory seromarkers, renal, cardiac and hepatic function tests were measured upon admission to Huoshenshan Hospital (Table 2). Of the 65 individuals, 59 patients had white blood cell counts in the normal range; eight patients had increased platelet counts; 30 patients had increased C-reactive protein

(CRP). Eleven out of 39 patients had increased procalcitonin (PCT). Almost all patients were normal regrading cardiac, renal and hepatic function tests. Thirty-five patients were tested for the serum antibodies to SARS-CoV-2, of which all patients were positive for SARS-CoV-2 IgG, while only 88.6% of patients were positive for SARS-CoV-2 IgM. In this study cohort, 26 out of 65 patients (40.0%) were treated with antiviral agents, including ribavirin, arbidol, oseltamivir and entecavir. Thirty-seven out of 65 patients (56.9%) were treated with antibiotics, which were mainly cephalosporin and floxacin. Notably, 95.4% of patients received traditional Chinese herbal tea or Chinese patent medicine, such as Lianhua Qingwen Capsule. Corticosteroid was mainly provided to the severe or critical cases. Twenty-six out of 65 patients had poor sleep quality and required prescription of hypnotics. The median DoVC in the overall study cohort was 48 days (ranged from 21 to 104 days). More than a half of the patients (53.8%) had a DoVC in the range of 31 to 60 days, and the DoVC was longer than 2 months in 30.8% of patients.

### Risk factors associated with longer viral clearance

In univariate analyses, several clinicopathologic characteristics were found to be associated with the DoVC, including fever ($p = 0.03$), shortness of breath ($p = 0.03$), and fatigue ($p = 0.03$) at onset, as well as serum levels of CRP ($p = 0.01$) and SARS-CoV-2 IgG ($p = 0.01$) (Tables 1 and 2). Thirty-five patients with complete data were included in the Cox model with a -2 Log Likelihood being 147.9 (overall Chi-square value = 27.909, $p < 0.001$), indicating an acceptable goodness-of-fit of the current model. Sensitivity analyses suggested no statistically significant difference in baseline and clinical characteristics between overall study cohort and patients included in Cox model (see Tables S1, S2). Multivariate analysis suggested that fever at illness onset [Hazard ratio (HR) = 4.897, 95% CI [1.809–13.260], $p = 0.002$], serum level of AST > 21.8 IU/L (HR = 3.010, 95% CI [1.195–7.578], $p = 0.019$), and titer of SARS-CoV-2 IgG > 142.09 AU/ml (HR = 3.061, 95% CI [1.263–7.415], $p = 0.013$) were the three independent risk factors associated with longer viral clearance in the refractory COVID-19 patients (Table 3). The proportional curves of positive viral test specified by three independent factors were depicted using Kaplan–Meier method (Fig. 1). The estimated median DoVC was significantly longer in patients with fever at admission (54 days *vs.* 44 days, $p = 0.028$), with a titer of SARS-CoV-2 IgG higher than 142 AU/ml (68 days *vs.* 56 days, $p = 0.013$). Patients with a serum level of AST > 21.8 IU/L were deemed to have only a statistically boundary significant longer DoVC (56 days *vs.* 44 days, $p = 0.089$).

## DISCUSSION

The DoVC is central for decision-making of nosocomial isolation precaution and post-discharge quarantine. Due to the high infectivity of SARS-CoV-2 and the overloaded medical system, a refractory clinical course of COVID-19 could be rather annoying for both patients and medical workers. However, there is still a lack of clinical guideline to deal with this particular group of patients. The current study documented the demographical, epidemiological, laboratory, and many other clinical data in a cohort of patients with a refractory course of COVID-19, with the purpose to identify the risk factors of prolonged

**Table 2  Clinicopathologic characteristics of patients with a refractory clinical course of COVID-19.**

| Clinicopathologic characteristics | Study cohort = 65 frequency (%)/Median (Q1, Q3) | Normal range | p value[*] |
|---|---|---|---|
| **Laboratory results** | | | |
| WBC # ($\times 10^9$/L) | 6.00 (5.05, 7.10) | 3.5–9.5 | 0.90 |
| RBC # ($\times 10^{12}$/L) | 4.10 (3.80, 4.45) | 4.3–5.8 | 0.76 |
| Platelet # ($\times 10^9$/L) | 230.00 (184.5, 291.50) | 125–350 | **0.07** |
| NLR | 2.35 (1.77, 3.45) | | **0.08** |
| C-reactive protein (mg/L) | 2.90 (1.59, 9.66) | 0–4 | **0.01** |
| Normal | 35 (53.8) | | |
| Increased | 30 (46.2) | | |
| Procalcitonin (ng/ml) ($n = 38$) | 0.05 (0.03, 0.06) | 0–0.05 | 0.12 |
| Normal | 27 (71.1) | | |
| Increased | 11 (28.9) | | |
| Albumin (g/L) | 38.15 (35.35, 40.65) | 40–55 | 0.54 |
| ALT (IU/L) | 30.15 (16.60, 49.85) | 9–50 | 0.23 |
| AST (IU/L) | 21.80 (17.25, 29.20) | 9–60 | **0.09** |
| HBDH (IU/L) ($n = 48$) | 152.90 (128.20, 193.43) | 24–190 | 0.33 |
| LDH (IU/L) ($n = 48$) | 181.15 (156.83, 239.95) | 120–250 | 0.44 |
| CK (IU/L) ($n = 48$) | 48.05 (34.35, 70.43) | 24–190 | 0.74 |
| CK-MB (IU/L) ($n = 48$) | 7.75 (6.40, 11.50) | 0–24 | 0.96 |
| SARS-CoV-2 IgM ($n = 35$) (AU/ml) | 39.01 (17.74, 90.51) | <10 | 0.26 |
| SARS-CoV-2 IgG ($n = 35$) (AU/ml) | 142.09 (116.40,179.86) | <10 | **0.01** |
| **Treatment details** | | | |
| Antivirals | 26 (40.0) | — | 0.12 |
| Antibiotics | 37 (56.9) | — | 0.24 |
| Traditional Chinese medicine | 62 (95.4) | — | 0.11 |
| Corticosteroid | 12 (18.5) | — | 0.27 |
| Hypnotics | 26 (40.0) | — | 0.66 |
| **Duration of viral clearance** | 49 (35, 65) | — | — |
| 21–30 days | 10 (15.4) | | |
| 31–60 days | 35 (53.8) | | |
| >60 days | 20 (30.8) | | |

Notes.
[*]Univariate analysis to determine the association of clinicopathologic characteristics with duration of viral clearance by using log-rank test. Patients were categorized into two groups based on the median value of laboratory results except for C-reactive protein and procalcitonin. NLR, neutrophil-lymphocyte ratio; ALT, Alanine aminotransferase; AST, Aspartate aminotransferase; HBDH, α-Hydroxybutyrate dehydrogenase; LDH, lactase dehydrogenase; CK, creatine kinase; CK-MB, creatine kinase isoenzyme. Variables with p value in bold were included in multivariable analysis.

hospitalization or viral clearance in these patients. In this retrospective study, 11 out of 65 (16.9%) were considered as possible reinfection (Table 1), given that they were negative for viral RNA in the former two consecutive nucleic acid tests. However, there was no concrete evidence to support this assumption as the false negative rate of nucleic acid testing was too high. Therefore, they were collectively recognized as patients with a refractory clinical course of COVID-19, along with those with recrudescent or persistent symptoms. It should be noted that this definition did not necessarily signify a severe or critical disease.

**Table 3  Multivariate analysis using Cox regression model ($n = 35$).**

| Variables (reference) | Hazard ratio | 95% confidence interval | | p value |
|---|---|---|---|---|
| | | Lower | Upper | |
| Fever (No) | | | | |
| Yes | 4.897 | 1.809 | 13.260 | **0.002** |
| Shortness of breath (No) | | | | |
| Yes | 1.239 | 0.436 | 3.521 | 0.687 |
| Fatigue (No) | | | | |
| Yes | 1.515 | 0.569 | 4.034 | 0.406 |
| Platelet # ($\leq 230 \times 10^9$/L) | | | | |
| $>230 \times 10^9$/L | 0.479 | 0.205 | 1.123 | 0.091 |
| NLR ($\leq 2.35$) | | | | |
| $>2.35$ | 2.831 | 0.984 | 8.145 | 0.054 |
| CRP ($\leq 4$ mg/L) | | | | |
| $>4$ mg/L | 0.311 | 0.087 | 1.115 | 0.073 |
| AST ($\leq 21.8$ IU/L) | | | | |
| $>21.8$ IU/L | 3.010 | 1.195 | 7.578 | **0.019** |
| SARS-CoV-2 IgG ($\leq 142.09$ ) | | | | |
| $>142.09$ | 3.061 | 1.263 | 7.415 | **0.013** |

**Notes.**

NLR, neutrophil-lymphocyte ratio; CRP, C-reactive protein; AST, aspartate aminotransferase; SARS-CoV-2 IgG, Severe acute respiratory syndrome coronavirus 2 immunoglobulin G.

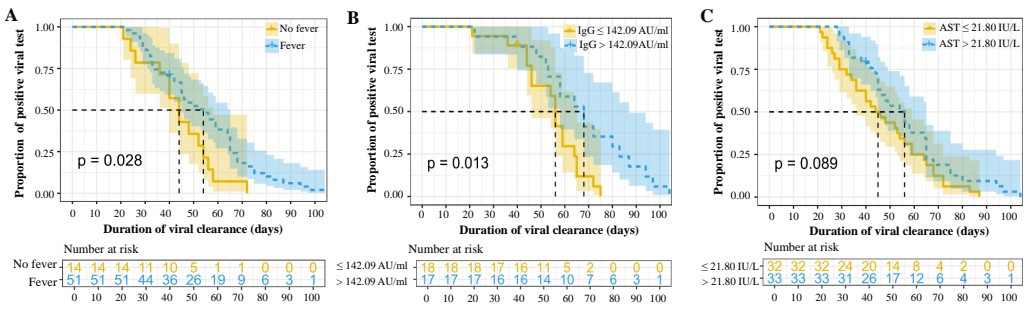

**Figure 1  Proportional curves of positive viral testing (A-C).**

Before the cross-study comparison, it is worth noting that the definition of duration of viral shedding or viral clearance has not been unified in the previous studies (*Yan et al., 2020*; *Xu et al., 2020*; *Hu et al., 2020*). In consideration of the high false negative rate of nucleic acid testing, we adopted a more conservative criteria to define the date of discharge as the endpoint of viral clearance. As two patients were still in hospitalization due to the complications from other organ systems despite their negative viral RNA results, their DoVC was calculated from the date of illness onset to the date of the second negative nucleic acid testing. The median DoVC was 48 days in our study, which was similar to the finding of one previous study (median = 53.5 days, IQR 47.75–60.5 days) (*Li, Wang & Lv, 2020*). The duration of viral shedding was about 20 days in many other studies, which

was much shorter because the patients with multiple COVID-19-related admissions were usually not enrolled (*Zhou et al., 2020*; *Yan et al., 2020*; *Qi et al., 2020*; *Fu et al., 2020*; *Xu et al., 2020*). It should be noted that nasal or pharyngeal swabs were mostly used for nucleic testing in these studies, but persistent SARS-CoV-2 RNA has been found elsewhere (*e.g.*, anal swab or stool sample) after negative conversion of nasopharyngeal RT-PCR test (*Wang et al., 2020*; *Kipkorir et al., 2020*).

In the current study, the DoVC had a very wide range (Table 2), which indicated that the further stratification among this population might be helpful for personalized management. We have identified the presence of fever at illness onset, AST > 21.8 IU/L and titer of SARS-CoV-2 IgG > 142.09 AU/ml to be the three independent risk factors for longer viral clearance in the refractory COVID-19 patients (Table 3, Fig. 1). Several other studies have also revealed the presence of fever was significantly associated with prolonged DoVC (*Zhou et al., 2021*; *Bennasrallah et al., 2021*). These patients were hypothesized to be more severely affected by the SARS-CoV-2 infection in the lungs and thus the prolonged DoVC (*Zhou et al., 2021*). Fever at illness onset may indicate a cytokine storm or a higher viral load at the time of symptom onset (*Chau et al., 2021*), as the viral load was found to be associated with more severe COVID-19 and a longer virus-shedding period (*Liu et al., 2020b*). Mildly abnormal liver function test was frequently reported in COVID-19 patients, especially the elevated plasma AST (*Kipkorir et al., 2020*; *Liu et al., 2020b*; *Bertolini et al., 2020*; *Zarifian et al., 2020*). Although a serum level of AST > 21.8 IU/L in our study not necessarily signified clinical relevance, it was significantly associated with a delayed viral clearance in multivariable analysis. Drug-induced liver injury should be taken into consideration (*Bertolini et al., 2020*), because a longer hospital stay usually required more medication, which were potentially hepatotoxic. Other possible underlying mechanisms included direct viral hepatitis or immune-mediated inflammatory response (*Zhou et al., 2021*). The presence of angiotensin-converting enzyme 2 (ACE2) receptor on cholangiocytes and hepatocytes suggested a plausible mechanism of SARS-CoV-2 related hepatotoxicity (*Chai et al., 2020*). Interestingly, a high level of SARS-CoV-2 IgM was also found to be related to the prolonged viral shedding in a previous study (*Li, Wang & Lv, 2020*), which was instead echoed by the level of IgG in our study. The larger magnitude of SARS-CoV-2 IgG, along with presence of fever at illness onset, might imply a higher viral load or stronger immune response to the virus. Nevertheless, the underlying mechanism requires further investigation.

Although less importantly, attention should also be paid to the symptoms such as shortness of breath, fatigue and the serum level of CRP (Tables 1 and 2). However, we did not observe the impact of age, comorbidities, severity of disease and treatment on the DoVC. Previous studies revealed multiple risk factors associated with prolonged viral shedding, including age (*Yan et al., 2020*; *Hu et al., 2020*), sex (*Xu et al., 2020*), fever (*Qi et al., 2020*; *Lu et al., 2020*) or chest tightness (*Hu et al., 2020*) at admission, comorbidities such as hypertension and diabetes mellitus (*Zhou et al., 2021*), the duration from symptom onset to admission (*Qi et al., 2020*; *Xu et al., 2020*), albumin level (*Fu et al., 2020*), antiviral treatment (*Yan et al., 2020*; *Fu et al., 2020*), invasive mechanical ventilation (*Xu et al., 2020*), and length of hospital stay (*Qi et al., 2020*). What is noteworthy is that many studies

directly divided the patients into two groups using a cutoff value in the DoVC and perform multivariate analysis by logistic regression, instead of using Kaplan–Meier method and Cox regression model (*Yan et al., 2020*; *Qi et al., 2020*; *Xu et al., 2020*). This might have led to a loss of detailed temporal information, and thus the inaccuracy of results.

Although this study is bolstered by its design and methodological strength, it is also limited by its retrospective nature and small sample size. Not all tests were performed and monitored in all patients during hospitalization, especially the serology testing for SARS-CoV-2. Therefore, patients with incomplete data were excluded from multivariable regression analysis, which limited its statistical power despite a sensitivity analysis had proved the representativeness of the included samples. The patients with multiple hospitalizations might potentially subject to recall bias in retrieving the treatment history. Additionally, we only enrolled patients with refractory COVID-19, therefore, the results and conclusion should be customized to this specific group of patients, which limits the generalizability of this study. Finally, the ability to associate clinicopathologic characteristics with DoVC might be limited by using a convenient instead of statistically determined sample size.

In conclusion, clinical characteristics such as fever at illness onset and a high serum level of AST or SARS-CoV-2 IgG were associated with delayed viral clearance in patients with a refractory course of COVID-19. Patients with these characteristics might need a more individualized treatment strategy to accelerate their recovery from the refractory COVID-19.

## ACKNOWLEDGEMENTS

We thank all the patients who provided consents to participate in this retrospective study.

### Funding
This study was funded by the 2020 Guangdong Provincial Special Project for Popularization of Science and Technology Innovation (grant number: 2020A1414070007). The funders had no role in study design, data collection and analysis, decision to publish, or preparation of the manuscript.

### Grant Disclosures
The following grant information was disclosed by the authors:
2020 Guangdong Provincial Special Project for Popularization of Science and Technology Innovation:  2020A1414070007.

### Competing Interests
The authors declare there are no competing interests.

## Author Contributions

- Weitao Zhuang and Shujie Huang conceived and designed the experiments, analyzed the data, prepared figures and/or tables, authored or reviewed drafts of the paper, and approved the final draft.
- Dongya Wang and Lulu Zha performed the experiments, authored or reviewed drafts of the paper, and approved the final draft.
- Wei Xu analyzed the data, authored or reviewed drafts of the paper, and approved the final draft.
- Guibin Qiao conceived and designed the experiments, authored or reviewed drafts of the paper, and approved the final draft.

## Human Ethics

The following information was supplied relating to ethical approvals (i.e., approving body and any reference numbers):

This study was approved by the institutional review boards of Huoshenshan Hospital (No. IEC-AF-SL-02), Guangdong Provincial People's Hospital (No. GDRE20200141H) and General Hospital of Southern Theater Command (No. 202039).

## Data Availability

The desensitized raw clinical or laboratory data used for analysis in this article are available in the Supplementary Files.

## Supplemental Information

Supplemental information for this article can be found online at http://dx.doi.org/10.7717/peerj.12535#supplemental-information.

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
