# Peer review of "Risk factors associated with prolonged viral clearance in patients with a refractory course of COVID-19: a retrospective study"

_PeerJ, doi:10.7717/peerj.12535_

## Round 0.1 · original submission · Major Revisions

I have taken over handling this submission as the original Academic Editor is no longer available.

Two reviewers have written their reports highlighting several issues which you should address in a new revised version of the text.

·

Basic reporting

See below

Experimental design

See below

Validity of the findings

See below

Additional comments

Dear authors and editors,
Thank you for the opportunity to review this paper. In this study, the authors reported the viral clearance in COVID-19 patients with refractory course and the risk factors. Since reinfection, relapsed infection and recurrence of positive (re-positive) nucleic acid detection are not rare in clinical practice and might have clinical and epidemiological implications for treatment and infection control measures. Thus, this topic is important, and relevant research to understand the underling mechanism and the risk factors are urgently needed. However, there are several significant weaknesses in this paper. My comments regarding this manuscript are provider below:
Major points:
1. According to the description of the manuscript, it seems that the study population was COVID-19 patients with refractory course rather than those with readmission. To the best of my knowledge, the reason that patients were admitted to the Huoshenshan hospital was not necessary to be readmission. Some of them were did not discharge from another hospital previously and were directly transferred to the hospital due to the absence of ward beds on elsewhere. Thus, the term “patients with more than one COVID-19 related hospitalization” means nothing, and the title is not accordance with the contents of the manuscript. I suggest the authors to revise the title correspondingly.
2. Given the tremendous body of literature demonstrating the viral clearance in COVID-19 patients with refractory course and the risk factors, it is not fully clear what the authors wish to add to the literature in the manuscript. The novelty of findings from this cohort seems to be limited.
3. The authors intended to explore the risk factors associated with the duration of viral clearance. However, the definition of the duration of viral clearance was the period from the date of illness onset (as stated in the history of present illness) to the date of discharge from Huoshenshan hospital. This definition was less accurate. Some patients might still be still hospitalization due to the complications from other organs (e.g. bacterial pneumonia, urinary tract infection, heart failure, etc.), even the negative results of SARS-Cov-2 RNA were received. This situation is more common in severe or critical patients. Thus, the research question is more about the duration of prolonged hospitalization instead of viral clearance. Please reflect on this.
4. It is noted that there were missing data in the antibodies of SARS-Cov-2 and PCT. However, no statement on the degree to which their variables had missing data and what was done in the cases of missing data or inconclusive data. In particular, how the incomplete data of antibodies were included into the COX model with other complete data? Are there any statistical methods performed? (e.g. imputation?)
5. The writing of this manuscript is poor. I suggest revising the introduction to make it succinct, so that that the rationale is clearer. Moreover, the discussion is very higgledy-piggledy, and the key points are difficult to grasp. Particularly, the authors did not show how their results and interpretations agree (or contrast) with previously published work.
6. The English language should be improved to ensure that an international audience can clearly understand your text. The current phrasing makes comprehension difficult. I suggest you have a colleague who is proficient in English and familiar with the subject matter review your manuscript, or read a textbook on how to write an academic article (e.g. Robert A. Day. How to write and published a scientific paper) before rewriting.
Minor points:
1. Line 43: “has” rather than “have”
2. Line 64: This paper did not focus on the underlying mechanism, and therefore the statement should be revised.
3. Line 126: The symptoms were presented from the illness onset or on this admission?
4. Line 132: 60+9=68?
5. Line 73: There was no definition of critical case, whereas critical cases were presented in the results.
6. Line 165: To the best of my knowledge, this is not the first study that reported the viral clearance in COVID-19 patients with refractory course and the risk factors.
7. The limitation of this study was not fully presented.

·

Basic reporting

no comment

Experimental design

no comment

Validity of the findings

no comment

Additional comments

The authors of the MS titled with “Risk factors associated with longer viral clearance in patients with more than one COVID-19-related hospitalizations”, firstly investigated the viral clearance in patients with a refractory course of COVID-19. They performed the time-dependent cox regression and found that fever at illness onset, serum level of aspartate aminotransferase > 21.7 IU/L, and titer of SARS-CoV-2 IgG > 141.69 were the three independent risk factors associated with delayed viral clearance. The study is novelty, had clinical importance, used comprehensive analyses, and the findings are of important value in management of COVID-19 patients.

Comments:
1. How was the sample size determined? Please add statistical analyses details of how the sample size was determined.
2. Whether the discharge criteria were same or not between the first and second COVID-19-related hospitalization?
3. Authors should further discuss the underlying mechanisms or some related factors for the associations among fever at illness onset, serum level of aspartate aminotransferase, titer of SARS-CoV-2 IgG and the delayed viral clearance.
4. The MS was well written.

---

## Round 0.2 · Major Revisions

Still pending some major issues which you should address in a new revised version of the text.

·

Basic reporting

see below

Experimental design

see below

Validity of the findings

see below

Additional comments

Dear editors and authors
Thank you for addressing the majority of my concerns. The quality of the manuscript has been improved significantly. However, there are still some issues that should be further addressed.
#1. Commonly, readmission is the act of taking somebody into a hospital again after they had been allowed to leave. As 78% of the COVID-19 patients were admitted to the Huoshengshan Hospital due to the persistent symptoms, the term “readmission” is not suitable for the study population. Please revise it across the manuscript. In addition, is the presentation “more than one COVID-19 related hospitalization” suitable? Please reflect on this.
#2. In the revised manuscript, the title as well as relevant presentation was revised to be “patients with a refractory course of COVID-19”. What is the exact definition of refractory course? No definition was presented in the manuscript. I have noticed that there were COVID-19 patients who had the duration of viral clearance with only 11 days. Did these patients meet the definition of refractory course? Please consider carefully how to describe the study population exactly.
#3 As the authors stated, there were 2 patients who were still hospitalization due to the complications from other organs, even the negative results of SARS-Cov-2 RNA were received. The definition of the duration of viral clearance the authors defined is not rigorous in case that the two patients were excluded. Whether the inclusion of the two patients would influence the results is still unknown (underestimate, overestimate, or change the result?). This problem can not be resolved by presenting it as a limitation. In this case, it is better to exclude the two patients, or only their durations from illness onset to viral shedding could be analyzed in statistical analysis. Or else, the research question is more about the duration of prolonged hospitalization instead of viral clearance.
#4. To deal with the missing data, the authors delete all the cases that had missing values. The sensitive analysis is nice. However, as cases with missing data were excluded from the COX model, there were only 36 cases in the model. Since the sample size may not be sufficient, the goodness-of-fit of the COX model should be reported. Additionally, the results should be interpreted carefully.
#5. The presentation of the conclusion in the abstract is not suitable. Commonly, this section is limited to the major findings based on the current study rather than the perspectives from the findings.
#6. Additionally, some minor work needed is grammatical errors throughout. Some examples include:
Line 197: …as the viral load was found to (be) associated with more severe COVID-19 and (a) longer virus-shedding period…
Please check the grammatical issues carefully across the manuscript.

---

## Round 0.3 · Minor Revisions

Still pending some minor issues to be addressed in a new version of the text. Please, see the reviewer's comments below.

·

Basic reporting

See below

Experimental design

OK

Validity of the findings

OK

Additional comments

Dear authors,
Thank you for addressing all of my concerns. The quality of the manuscript has been improved more significantly. As a scientific article, a clear, unambiguous, professional presentation is needed throughout. However, there are still several minor issues to be corrected, majority of which are regarding professional presentation.
1. Professional presentations regarding the medical nomenclature
For example:
#line 35: The presentation “mildly impaired liver function” is not correct. As the upper limit range of AST Is 60 IU/L, a level of AST> 21.8 IU/L does not mean mild impaired liver function.
#line 37: What does the “COVID-19 diseases” mean? COVID-19 means Coronavirus disease 2019.
#line 43:The presentation “ COVID-19 infection” is inaccurate. Maybe “SARS-Cov-2 infection” is more precise.
#line 141: As I have point out previously, the presentation “readmitted” is not correct.
#Table 2: “20-30 days”. Maybe “21-30 days” will be more precise because that the duration of 20 days was excluded.
2. There are also several ambiguous presentations across the manuscript. As an academic writer with English as a second language, I strongly recommend the authors to read a textbook regarding the scientific writing named “English Exposed-Common Mistakes Made by Chinese Speakers” (Steve Hart, Hongkong University Press).
For example:
#line 36 What is the exact mean of “positive treatment”? Is the presentation “positive” suitable here?
#line 60: What is the exact mean of “diseases”? Are the transmission of any other diseases was also involved?
#Line 141:”hospitals”. Did the included patients be admitted in other hospitals?
#Line 172, 177, 208, 355……: The exact titer of SARS-Cov-2 IgG should be 142.09 AU/ml. The unit should not be missed. Please re-check it across the manuscript.
#line 184 “virus”. Which virus do you mention in this sentence?
#Line 199: was was…..
3. Others
#line 217: Commonly, the exact statistical values in the result are not recommended to be presented in the discussion repeatedly.

---

## Round 0.4 · accepted · Accept

All the reviewers' concerns have been correctly addressed by the authors, therefore I am pleased to tell you that your paper has been accepted for publication in PeerJ. Congratulations!

·

Basic reporting

OK

Experimental design

OK

Validity of the findings

OK

Additional comments

The authors have addressed all my concerns. Good luck!